# Research on Surface Defect Detection of Strip Steel Based on Improved YOLOv7

**DOI:** 10.3390/s24092667

**Published:** 2024-04-23

**Authors:** Baozhan Lv, Beiyang Duan, Yeming Zhang, Shuping Li, Feng Wei, Sanpeng Gong, Qiji Ma, Maolin Cai

**Affiliations:** 1School of Mechanical and Power Engineering, Henan Polytechnic University, Jiaozuo 454003, China; baozhan@hpu.edu.cn (B.L.); 212105020060@home.hpu.edu.cn (B.D.); lishuping@hpu.edu.cn (S.L.); elite@hpu.edu.cn (F.W.); gongsp@hpu.edu.cn (S.G.); 212305010031@home.hpu.edu.cn (Q.M.); 2School of Automation Science and Electrical Engineering, Beihang University, Beijing 100191, China; caimaolin@buaa.edu.cn

**Keywords:** defect detection, attention mechanism, YOLOv7, PConv, SPD

## Abstract

Surface defect detection of strip steel is an important guarantee for improving the production quality of strip steel. However, due to the diverse types, scales, and texture structures of surface defects on strip steel, as well as the irregular distribution of defects, it is difficult to achieve rapid and accurate detection of strip steel surface defects with existing methods. This article proposes a real-time and high-precision surface defect detection algorithm for strip steel based on YOLOv7. Firstly, Partial Conv is used to replace the conventional convolution blocks of the backbone network to reduce the size of the network model and improve the speed of detection; Secondly, The CA attention mechanism module is added to the ELAN module to enhance the ability of the network to extract detect features and improve the effectiveness of detect detection in complex environments; Finally, The SPD convolution module is introduced at the output end to improve the detection performance of small targets with surface defects on steel. The experimental results on the NEU-DET dataset indicate that the mean average accuracy (mAP@IoU = 0.5) is 80.4%, which is 4.0% higher than the baseline network. The number of parameters is reduced by 8.9%, and the computational load is reduced by 21.9% (GFLOPs). The detection speed reaches 90.9 FPS, which can well meet the requirements of real-time detection.

## 1. Introduction

Strip steel is one of the core products in the steel industry and has become an important raw material in industries such as automotive, mechanical manufacturing, chemical equipment, and aerospace. With the booming development of high-end industries such as aerospace, automotive, and precision machinery manufacturing, the industry has put forward higher requirements for the quality of strip steel products. However, the production process of strip steel is inevitably affected by various factors, resulting in defects such as scratches, cracks, and oxidation on its surface, seriously affecting the production efficiency and product quality of strip steel. Therefore, improving the ability to detect surface defects on strip steel and helping to detect defective products in the production process early is of great practical significance for improving product quality and improving work efficiency [1,2].

In recent years, domestic and foreign scholars have conducted extensive research on defect detection in computer vision technology, with two main research methods: machine learning and deep learning. The surface defect detection technology of industrial products based on machine vision has become mature, mainly divided into four types of detection methods: statistical methods, spectral methods, model-based methods, and learning-based methods. Significant histogram features [3] and local binary patterns [4] are popular techniques in statistical methods, but these methods have obvious drawbacks: they often require defect features to be strength-separable and highly sensitive to noise. Spectral methods use Fourier transform [5], wavelet transform [6] and Gabor filter [7] to transform signals from the spatial domain to the frequency domain for defect recognition. The classic model-based methods in defect detection include Markov random field models [8] and autoregressive models [9], which are not satisfactory in terms of detection accuracy and are only suitable for defect detection in local images, often consuming more resources. Learning-based methods use support vector machines (SVMs) [10] and K-nearest neighbors (K-NNs) [11], which consider the statistical changes of defects in the image to discover the expected defects. One of the main drawbacks of this method is that it requires the development of precise models to discover patterns within defects, and they may still be less robust to changes in texture, lighting, complexity of defects, etc. The traditional machine vision methods mentioned above usually require a manual design to describe the characteristics of the defects. Therefore, based on human subjectivity, the characteristics of the manual design make it difficult to distinguish defects on industrial surfaces. Faced with unknown and diverse types of defects, detection methods often exhibit poor generalization ability. Therefore, when faced with more complex and irregular defects, traditional methods are difficult to apply in practical industrial application scenarios.

With the comprehensive intelligent development of the manufacturing industry, higher efficiency, shorter time consumption, higher accuracy, and lower cost requirements have been put forward for the defect detection of industrial products. Ultimately, surface defect detection based on deep learning has entered people’s vision. The methods based on deep learning mainly include one-stage YOLO [12], SSD [13], and two-stage Faster R-CNN [14], Mask R-CNN [15] algorithms. Two-stage object detection algorithms achieve target recognition through two core steps: Firstly, generating potential target regions (region proposals), a process which often involves extensive pre-screening operations such as using a region proposal network (RPN). Subsequently, these candidate regions undergo meticulous classification judgments and precise localization refinements. However, this segmented workflow inherently leads to compromises in computational efficiency and decreases in processing speed, attributes that render such algorithms less suitable for real-time demanding applications. Furthermore, two-stage detection methods require significantly higher system resources during operation, particularly when handling high-resolution images or large datasets, where memory usage can spike, creating a performance bottleneck. Moreover, their complex network architecture designs pose greater challenges in training and optimization, necessitating more time and computational resources compared to single-stage algorithms to attain optimal performance. In 2024, Fu et al. [16] developed an automatic detection and pixel-level quantification model based on the joint Mask R-CNN and TransUNet. The Mask RCNN model demonstrated an AP50 of 0.989 and AP75 of 0.864 for the image dataset of microcrack damage.

Compared to two-stage detection algorithms, one-stage algorithms are usually designed to be more concise and efficient. Through a single forward propagation, they can directly predict the category and corresponding bounding box coordinates of each position in the image, without the need to go through the process of selecting candidate regions for classification and regression, greatly improving processing speed and meeting the needs of real-time or high-speed scenes. In 2016, Redmon et al. [12] proposed an end-to-end object detection algorithm YOLOv1 (you only look once), which unified the object detection problem into a regression problem, ensuring a certain level of accuracy and speed in object detection. Subsequently, the Yolo series of algorithms were successively proposed, and corresponding progress was also made in the field of defect detection. In 2018, YOLOv3 [17], proposed by Redmon et al., borrowed the residual idea of ResNet, further improving its speed and accuracy. Zehao Zheng et al. [18] proposed an improved YOLOv3 model that includes a bottleneck attention network (BNA Net), an attention mechanism, a defect localization subnet, and a large-sized output feature branch. It achieved a 16.31% improvement compared to the original network on the bearing cover defect dataset, solving the problem of insensitivity of the original algorithm to medium- and large-sized targets. In 2021, Glenn Jocher et al. [19] proposed a new model YOLOv5, which uses k-means clustering to adaptively calculate anchor boxes during training. At the same time, the neck network adopts the CSP2 structure designed by CSPNet, further enhancing the network’s feature fusion ability. These methods significantly improve the detection speed of the network while maintaining detection accuracy. Jiacheng Fan et al. [20] proposed an ACD-YOLO model based on the YOLOv5 detection algorithm, which combines anchor box optimization, a context enhancement module, and an efficient convolution operator. The improved model achieved a 5.7% improvement in mAP on the NEU-DET strip defect dataset, reaching 79.3% and a frame rate of 72 FPS. Zhang et al. [21] proposed a novel SEM-based YOLOv5 model and combined it with the OMF segmentation algorithm for ceramic micro defect detection. The experimental results show that this method can effectively detect surface defects, namely defects and cracks, with a precision of 98% and an average detection time of 0.05 s. Zhang et al. [22] proposed an improved PP-YOLOE-m network to detect surface defects on strip steel. The improved network achieved an AP50 of 80.3% on the NEU-DET dataset and can run at a speed of 95 FPS on a single Tesla V100 GPU. Wang et al. [23] proposed the YOLOv7 algorithm, which effectively improves the detection efficiency of the algorithm through an efficient long-range aggregation network (ELAN) and a cascading-based model scaling strategy. However, missed detections are still inevitable in the process of detecting small target defect features. Gao et al. [24] proposed the CDN-YOLOv7 model based on the YOLOv7 algorithm. This model incorporates a CARAFE lightweight up-sampling operator, designs a detection head network that integrates the cascaded attention mechanism and decoupling head, and proposes NF-EIoU to replace the CIoU loss function in the original network based on the Focal EIoU loss function. The final mAP on the NEU-DET strip defect dataset reached 80.3%, with a frame rate of 73.4 FPS. From this, it can be seen that although there are various algorithms applied to strip the defect detection, most algorithms find it difficult to balance detection accuracy and detection speed. Therefore, researching high-precision real-time defect detection algorithms is of great practical significance.

This article addresses the problem of insufficient feature extraction ability and low model detection accuracy in current surface defect detection algorithms for steel strips. Based on the YOLOv7 series, the YOLOv7 algorithm is improved to improve the efficiency of steel surface defect detection. Firstly, a lightweight Partial Conv (PConv) [25] is used to replace some conventional convolutional blocks in the backbone network ELAN module, in order to reduce the size of the network model and improve the detection speed; Secondly, a Coordinate Attention (CA) [26] mechanism is added to the last convolution layer of the middle two ELAN modules to enhance the network’s ability to extract image features and improve the effectiveness of object detection in complex environments; Finally, an SPD convolution module [27] is introduced at the output end to improve the detection performance of small targets with surface defects on steel. The specific structure is shown in Figure 1. The improved YOLOv7 algorithm proposed in this article was tested on the NEU-DET dataset, and the experiments showed that the method has good detection performance in surface defect detection tasks of strip steel, which can further meet industrial deployment requirements.

## 2. Methodology

### 2.1. Baseline Networks

The YOLOv7 algorithm adopts the extended efficient long range attention network (E-ELAN), a cascaded model-based scaling and re-parameterized convolutional layer (REP-Conv) strategy, achieving a good balance between detection efficiency and accuracy. The YOLOv7 network structure consists of four modules: input, backbone, neck, and head. The input end uses Mosaic technology to improve the training speed and reduce the memory consumption. The image undergoes a series of preprocessing operations such as cropping and scaling at the input end to unify the pixels and meet the requirements of the feature extraction network. Backbone consists of modules such as CBS, E-ELAN, and MP, which are used to extract feature information of input objects. The neck section is mainly responsible for feature fusion, achieving the fusion of resolution and high semantic information. The head section consists of three detection heads, mainly responsible for achieving target prediction.

### 2.2. Partial Conv (PConv)

In the industrial production of strip steel, the defect characteristics require real-time detection and differentiation, which puts high demands on the running speed of the detection network. Introducing Partial Convolution (PConv) in the backbone network to replace the original convolution can reduce redundant calculations and memory access, and more effectively extract spatial features. PConv is different from the previous approach adopted by many scholars to improve the computational speed of neural networks by reducing computational complexity. PConv solves the problem of low computational speed (FLOPS) caused by frequent memory access by reducing computational redundancy and memory access, ensuring high FLOPS while reducing FLOPs. The working principle of PConv is shown in the Figure 2: applying conventional Conv on a part of the input channel for spatial feature extraction, while keeping the remaining channels unchanged. For continuous or regular memory access, the first or last continuous *x_i_* channel is considered as a representative of the entire feature map for calculation, and each filter slides on one *x_i_* channel. Generally, it is considered that the input and output feature maps have the same number of channels. So, the FLOPs of PConv are
(1)$$h\times w\times m^{2}\times x_{i}^{2}$$

With a typical partial ratio $r=\frac{x_{i}}{x}=1/4$, the FLOPs of a PConv is only 1/16 of a regular Conv. Besides, PConv has a smaller amount of memory access, i.e.,
(2)$$h\times w\times 2x_{i}+m^{2}\times x_{i}^{2}\approx h\times w\times 2x_{i}$$
which is only 1/4 of a regular Conv for *r* = 1/4.

### 2.3. Coordinate Attention (CA)

The detection ability of existing surface defect detection algorithms is limited when facing complex and diverse surface defects of strip steel. At the same time, the detection effect is easily affected by factors such as image background noise and the irregular distribution of defect features, resulting in insufficient learning of surface defect features of strip steel by the detection network and making it difficult to obtain accurate defect feature positions. Therefore, how to enhance the location information of defect features and improve the network’s attention to defects is also one of the problems. In recent years, attention mechanisms have developed rapidly due to their plug-and-play characteristics and the advantages of effectively improving network detection performance, and have also been widely applied in the field of image defect detection. Here, we choose to introduce a coordinate attention mechanism (CA) at the last convolution of the middle two ELAN modules in the backbone network and the SPPCSPC module in the feature fusion layer. The specific structure of the CA module is shown in Figure 3. Compared to other types of channel attention mechanisms, it decomposes channel attention into two one-dimensional feature encoding processes, aggregating features along two spatial directions. Through this approach, precise positional information can be retained along one spatial direction, while long-distance dependencies can be captured along another spatial direction. The specific operations are divided into coordinate information embedding and coordinate attention generation. Therefore, the introduced network retains both the location information of defects and further enhances the feature information of defect features.

The structure of the CA module can be defined as:(3)$$y_{c}=\frac{1}{H\times W}\sum_{i=1}^{H}\sum_{j=1}^{W}x_{c}(i,j)$$

$y_{c}$ is the output associated with the *c*-th channel, x is the input convolutional layer, and $x_{c}\left(i,j\right)\;$is the position x of the $\left(i,j\right)$ input in the *c*-th channel convolutional layer. In attention mechanisms, global pooling is commonly used to globally encode spatial information, compressing it into channel descriptors and making it difficult to preserve positional information, which is crucial for the spatial structure in visual detection tasks. In order to enable the attention module to capture remote spatial interactions with precise positional information, the CA attention mechanism decomposes global pooling into a pair of one-dimensional feature-encoding operations. Specifically, given the input *x*, we use the two spatial ranges (*H*, 1) or (1, *W*) of the pooling kernel to encode each channel along the horizontal and vertical coordinates, respectively. Therefore, the output of the *c*-th channel at height *H* can be formulated as:(4)$$y_{c}^{h}(h)=\frac{1}{W}\sum_{0\leq i<W}x_{c}(h,i)$$

Similarly, the output of the *c*-th channel at width *W* can be formulated as:(5)$$y_{c}^{w}(w)=\frac{1}{H}\sum_{0\leq i<H}x_{c}(j,W)$$

Coordinate attention generation utilizes the above two equations, while fully utilizing the captured positional information, to focus on the relationships between channels. The specific operation is to combine the two equations and send them to the shared 1 × 1 convolutional transformation function *F*1, obtaining:(6)$$f=\lambda (F1([z^{h},z^{w}]))$$

[$z^{h},z^{w}$] represents cascading operations along spatial dimensions, γ is a non-linear activation function, where $f\in R^{C/r\times \left(H+W\right)}\;$ is an intermediate feature map that encodes spatial information in both horizontal and vertical directions. To reduce the complexity of the model, an appropriate reduction ratio r is usually used to reduce the number of channels in f. Then, we split *f* into two independent tensors, $f^{h}\in R^{C/r\times H}$ and $f^{w}\in R^{C/r\times W},$ along the spatial dimension. The other two 1 × 1 convolutional transformations, $F_{h}$ and $F_{w}$, are used to transform $F_{h}$ and $F_{w}$ into tensors with the same number of channels as input *X*, respectively, to obtain:(7)$$g^{h}=\eta (F_{h}(f^{h}))$$
(8)$$g^{w}=\eta (F_{w}(f^{w}))$$

η is a sigmoid function. At this point, the CA module has completed both vertical and horizontal attention. The CA model formula is defined as:(9)$$y_{c}\left(i,j\right)=x_{c}\left(i,j\right)\times \eta_{c}^{h}\left(i\right)\times \eta_{c}^{w}\left(j\right)$$

It decomposes global pooling into a pair of one-dimensional feature-encoding operations. Then, two one-dimensional global pooling operations are performed to aggregate the input features into two independent directional perception feature maps along the vertical and horizontal directions. The long-range dependencies of the feature maps are dynamically captured through the transformation of features in space, and weights are assigned to the spatial positions of defect features to enhance the detection network’s attention to defects. This enables the detection network to more accurately locate objects of interest, thereby helping the entire model to better recognize defect features.

### 2.4. SPD

To improve the detection effect of small defects such as pitting, scratches, and plaques on the surface of steel, a convolutional building block SPD is introduced at the output end to detect low resolution and small objects. The SPD convolution building block consists of spatial to depth layers (SPD layers) and non-stepped convolution layers. The SPD layer utilizes image conversion technology to down-sample the original feature map into an intermediate feature map with feature discrimination information. Its working principle is shown in the following Figure 4:

Firstly, given any original feature map *X*, its sub feature maps $f_{x,y}$ are composed of all entries *X(i, j)*, where *i + x* and *i + y* can be divided by a scale factor, and each sub feature map is down-sampling according to the scale factor. As shown in the figure, when scale = 2, four sub feature maps, $f_{0,0},\;f_{1,0}$, $f_{0,1}$, $f_{1,1}$, can be obtained. The shape of each sub map is $\left(\frac{S}{2},\frac{S}{2},C_{1}\right)$, which is equivalent to double down-sampling the original feature map *X*. Subsequently, the obtained sub feature maps are concatenated along the channel dimension to obtain the intermediate feature map *X′*, where the spatial dimension of *X′* is reduced by twice and the channel dimension is increased by twice. At this point, SPD transforms the original feature map $X(S,S,C_{1})$ into an intermediate feature map $X^{\prime }(\frac{S}{scale},\frac{S}{scale},{scale}^{2}C_{1})$ with feature discrimination information. Finally, by adding a non-stride (stride = 1) convolutional layer with a *C_2_* filter, the intermediate feature layer $X^{\prime }(\frac{S}{scale},\frac{S}{scale},{scale}^{2}C_{1})$ is further transformed into the final feature layer $X^{\prime \prime }(\frac{S}{scale},\frac{S}{scale},{scale}^{2}C_{2})$ while preserving as much feature discrimination information as possible.

When adding the SPD module to the YOLOv7 defect detection network, the SPD convolutional layer first splits the small or low-resolution defect feature maps on the steel surface into sub feature maps, then concatenates the sub feature maps into intermediate feature maps to extract feature identification information. Finally, the extracted feature identification information is filtered and learned through a filter. The above work makes the recognition of small targets and low-resolution defect features by the detection head more accurate, which can effectively improve the detection performance of the algorithm.

### 2.5. EIoU

In the original YOLOv7 baseline network, CIoU Loss [28] is used as the bounding box loss function, and its expression is as follows:(10)$$L_{CIoU}=1-IoU+\frac{\rho^{2}\left(b,b_{gt}\right)}{c^{2}}+\alpha \upsilon$$
where *IoU* represents the ratio of the overlapping area of the predicted and target borders to the overall area occupied, while b and $b_{gt}$ represent the center points of the predicted and target borders. Respectively, *ρ* represents the Euclidean distance of the center point, and c represents the diagonal distance between the predicted bounding box and the minimum rectangle outside the target bounding box; $\upsilon =\frac{4}{\pi^{2}}{\left(\arctan\frac{w_{gt}}{h_{gt}}-\arctan\frac{w}{h}\right)}^{2}$, w,h, and $w_{gt}$, $h_{gt}$ respectively represent the predicted border and target border widths and heights, which are used to characterize the consistency of length and width. $\alpha =\frac{\upsilon }{\left(1-IoU\right)+\upsilon }$ is the regulatory factor. υ The gradient calculation related to u for *w* and *h* is as follows:(11)$$\frac{\partial \upsilon }{\partial w}=\frac{8}{\pi }\left(\arctan\frac{w^{gt}}{h^{gt}}-\arctan\frac{w}{h}\right)\times \frac{h}{w^{2}+h^{2}}$$
(12)$$\frac{\partial \upsilon }{\partial h}=-\frac{8}{\pi }\left(\arctan\frac{w^{gt}}{h^{gt}}-\arctan\frac{w}{h}\right)\times \frac{w}{w^{2}+h^{2}}$$

According to Formulas (11) and (12), it can be inferred that the $\frac{\partial \upsilon }{\partial w}=-\frac{\partial \upsilon }{\partial h}\times \frac{h}{w}$ gradient sign is opposite. Therefore, both variables will inevitably increase and decrease during the optimization process. In addition, when the width and height of the prediction box satisfy $\left\{\left(w=kw^{gt},h=kh^{gt}\right)\left|k\in R^{+}\right.\right\}$, υ = 0, the relative width to height ratio of the supplementary item will lose its effect. Due to the above two factors, the convergence speed of the CIoU loss has slowed down.

In order to compensate for the shortcomings of the CIoU loss function, this paper replaces it with the EIoU loss function [29], which minimizes the width and height differences between the target box and the prediction box. This not only accelerates the convergence speed of the detection network training process, but also improves the accuracy of regression. The EIoU loss function formula is defined as:(13)$$L_{EIoU}=L_{IoU}+L_{dis}+L_{asp}=1-IoU+\frac{\rho^{2}\left(b,b^{gt}\right)}{c^{2}}+\frac{\rho^{2}\left(w,w^{gt}\right)}{{c_{w}}^{2}}+\frac{\rho^{2}\left(h,h^{gt}\right)}{{c_{h}}^{2}}$$
where $c_{w}$ and $c_{h}$ are the width and height of the minimum rectangle outside the predicted and target bounding boxes, respectively. The EIoU loss function enhances the network’s regression ability for defect positions by minimizing the difference in width and height between the target and predicted boxes, further improving the predictive performance of the detection network.

## 3. Results and Discussion

Training and testing on the NEU-DET strip defect dataset to verify the effectiveness of the improved algorithm.

### 3.1. Experimental Preparation

#### 3.1.1. Dataset

The NEU-DET steel defect dataset contains 1800 grayscale images, including 1440 in the training set, 180 in the testing set, and 180 in the validation set, all with a resolution of 200 × 200 pixels. According to the common surface defects of steel, they are divided into six categories: crazing (Cr), inclusion (In), patches (Pa), pitted surface (Ps), rolled in scale (Rs), and scratches (Sc). The defect characteristics are shown in Figure 5. We conducted ablation experiments and comparative experiments on this dataset to train and validate the effectiveness of the improved module and algorithm.

#### 3.1.2. Experimental Environment and Parameter Setting

The hardware configuration for the experiment is Intel Core i512400F@2.5 GHz (Intel Corporation, Santa Clara, CA, USA). The processor and graphics card are NVIDIA GeForce RTX 3070 8 GB (Nvidia Corporation, Santa Clara, CA, USA). The software environment is CUDA10 2 and cuDNN8 2.1. The operating system is Windows 11(Microsoft Corporation, Redmond, WA, USA). The network model is built based on the Python framework, with Python version 3.9 and Python version 1.12.1. In the experiment, the batch size was set to 8, the epoch was set to 200, and the learning rate was set to 0.005.

#### 3.1.3. Object Detection Evaluation

The experiment uses six evaluation indicators: mAP, precision, recall, FPS, Params, and FLOPs, which are introduced as follows:*mAP*: The average recognition accuracy of all categories is reflected, and the calculation formula is:
(14)$$mAP=\frac{1}{c}\sum_{i=1}^{c}AP_{i}$$

Among them, *c* represents the total number of categories in the image, *i* represents the number of detections, and *AP* represents the average recognition accuracy of a single category. mAP@0.5 refers to the average value obtained by adding the average recognition accuracy *AP* of each category when IoU is set to 0.5.Precision: It reflects the accuracy of model detection, calculated using the formula:


(15)
$$Precision=\frac{TP}{TP+FP}$$


where *TP* is the true case and *FP* is the false certificate case.Recall: It represents the proportion of correctly predicted positive examples:


(16)
$$Recall=\frac{TP}{TP+FN}$$


where *FN* represents data that were mistakenly identified by the model as negative examples but were actually positive examples.*FPS* represents the number of image frames processed within one second, and the calculation formula is:


(17)
$$FPS=\frac{1}{Processing\;time\;per\;frame}$$


where *Processing time per frame* represents the processing time of each frame, including the image preprocessing time, the model inference time, and the post-processing time.Params reflect the number of parameters occupied by the model’s memory.FLOPs reflect the computational complexity of the model.

### 3.2. Ablation Experiment

In order to verify the effectiveness of improving YOLOv7, this paper conducted progressive performance tests on each improvement point, including PConv, CA, and SPD modules, using YOLOv7 as the benchmark network. Table 1 shows the results of the ablation experiment. From the experimental results in Table 1, it can be seen that: firstly, the improved network model reduced the number of parameters by 12.1%, the computational complexity by 19.3%, the detection speed by 6.4 FPS, and the mAP of six types of defect features increased by 2.2% compared to the baseline network after using PConv convolution blocks instead of some conventional convolution modules in the backbone network. This is because the PConv introduced by the improved model effectively reduces redundant calculations and memory access in the process of extracting defect feature information. We enabled the model to fully utilize the computing power of hardware devices. Meanwhile, the reduction in parameter and computational complexity also makes the improved model easier to deploy in actual industrial production. Secondly, after introducing the CA attention mechanism block, the feature extraction ability of the backbone network was improved, and mAP was further improved by 0.7%. Finally, the SPD module introduced in the detection head further improved the recognition ability for small targets and low-resolution defects, with an mAP increase of 80.4%. At the same time, the detection speed can reach up to 90.9 FPS, which can well meet the requirements of real-time detection.

### 3.3. Comparative Experiment

#### 3.3.1. Comparison Experiment of Improvement Effect

In order to ensure fairness in the comparison of models, under the condition that all parameter settings remain unchanged, we trained both the original YOLOv7 algorithm and the improved version separately. The training results are depicted in Figure 6.

From the above data comparison, it can be seen that the improved YOLOv7 algorithm’s mAP value has increased from 76.4% to 80.4%, an increase of 4.0 percentage points. The AP value of Cr defects increased by 7.1 percentage points from 45.0% to 52.1%; The AP value of In defects increased by 2.6 percentage points from 86.1% to 88.7%; The AP value of the Pa defect increased by 0.4 percentage points from 94.2% to 94.6%; The AP value of the Ps defects increased by 4.6 percentage points from 91.1% to 95.7%; The AP value of the Rs defects increased by 7.3 percentage points from 54.7% to 62.0%; The AP value of the Sc defects increased by 2.3 percentage points from 86.9% to 89.2%, and the AP values detected for all six types of defects improved.

The comparison of improvement algorithms for six types of defects is shown in Figure 7. Various types of defect features are identified using rectangular boxes of different colors, and confidence is indicated in the upper left corner of each rectangular box. It can be seen that the improved YOLOv7 model has a positive impact on all six types of defects, especially on the detection of small and low-resolution target defect features, which reduces the risk of false positives and missed detections to a certain extent.

#### 3.3.2. Comparisons of Different Attention Mechanism Modules

To verify that the CA attention mechanism module selected in this article has the best detection performance, we used YOLOv7 as the baseline network and inserted SE [30], CBAM [31], and CA attention mechanism modules at the same position for comparison. The detection results of each module on the NEU-DET steel defect dataset are shown in Table 2. The comparison results show that the detection network using the SE module has the best detection performance on In and Pa types of defects, with AP values reaching 87.8% and 96.2%, respectively; The overall performance of the network using CBAM modules in detecting various defects is moderate. The network using the CA module has the best detection performance for Cr and Sc defects, with AP values of 49.7% and 91.2%, respectively. This fully demonstrates that adding the attention mechanism module to the YOLOv7 network for defect detection is an effective solution. Compared with the original YOLOv7 network, the detection network using the CA module significantly improved the detection performance of Cr, Rs, and Sc defects, with an AP improvement of 4.7%, 2.9%, and 4.3%, respectively. Compared with the other two networks using the SE and CBAM attention mechanism modules, the network using the CA module also achieved the best detection performance, with a highest mAP value of 78.1%. This is because SE only considers attention in the channel dimension and lacks the acquisition of defect feature information in the spatial dimension. The CBAM attention mechanism introduces positional information through global pooling on the channel, but this approach can only capture local information and cannot obtain long-range dependent information. On the other hand, the CA attention mechanism decomposes channel attention into vertical and horizontal directions, effectively integrating spatial coordinate information into the generated attention map. Then, we aggregated them into two separate directional perception feature maps. This approach can fully preserve the integrity of feature map position information and dynamically assign weights to the spatial positions of defect features, effectively improving the utilization of spatial defect information and the attention of the detection network to defects.

#### 3.3.3. Comparisons of Different IoU Loss Functions

To verify the effectiveness of the EIoU loss function used in this article in strip defect detection, CIoU, SIoU, and WIoU were compared on the improved network. The specific results are shown in Table 3. CIoU is a loss function used in the YOLOv7 original network, which has a maximum AP value of 62.9% when detecting Rs defects, but performs poorly in detecting Sc defects: only 85.5%. The overall detection performance is the worst among the four types of loss functions, with an mAP value of 79.3%. The detection performance of SIoU on Cr defects reached the highest AP value of 59.1%, but its detection performance on Ps and Rs defects was poor, at 89.8% and 58.3%, respectively. The performance of the WIoU loss function is relatively balanced, with a performance of 79.5% on mAP. Our work performed the best when using EIoU, and compared to the CIoU loss function used in the baseline network, the overall mAP value increased by 1.1%, to 80.4%. We achieved an improvement in AP in the detection of In, Pa, Ps, and Sc defects, which were 0.7%, 1.0%, 2.1%, and 3.7%, respectively. This is because CIoU did not consider the true difference between the anchor box width and height and their confidence, which affected the network’s localization of surface defect feature positions on the strip steel, resulting in the network being unable to capture complete defect features. However, the EIoU can effectively avoid this problem in this article, which enhances the network’s extraction of the spatial position information of surface defect features on the strip steel, and thus achieves the best detection performance.

## 4. Conclusions

A high-precision real-time defect detection algorithm based on YOLOv7 is proposed to address the issue of low accuracy in the surface defect detection of strip steel. This improved algorithm replaces the original convolution module with PConv convolution blocks in the backbone network, which not only reduces the model’s parameter and computational complexity, but also improves the detection accuracy and speed. It introduces the CA coordinate attention mechanism to enhance the network’s ability to extract image features. The use of SPD convolution modules at the output end improves the detection effect on small defects. The experimental results show that the detection speed of the improved algorithm is 90.9 FPS, and the mAP is 80.4%, proving that the improved algorithm can demonstrate good comprehensive performance in the surface defect detection of strip steel. Although the improved algorithm proposed in this article has achieved certain improvements in the accuracy of strip defect detection, its performance in dealing with complex texture defect features is still unsatisfactory, and a large amount of background noise seriously affects the detection effect. Therefore, further improving the detection accuracy of the model should still be the focus of subsequent research. In addition, although the parameter quantity of the improved network proposed in this article is reduced by 8.9% compared to the baseline network, due to the large number of parameters in YOLOv7 itself, further reducing the model’s parameter count to enable its deployment in actual production remains a top priority for subsequent research.

## Figures and Tables

**Figure 1 sensors-24-02667-f001:**
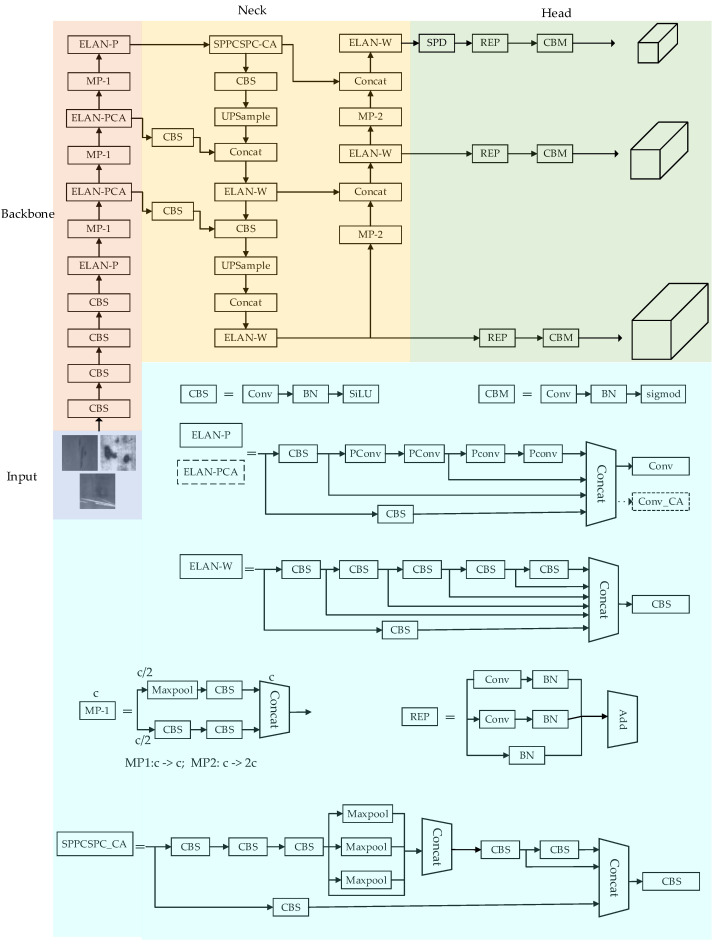
Improved YOLOv7 network structure.

**Figure 2 sensors-24-02667-f002:**
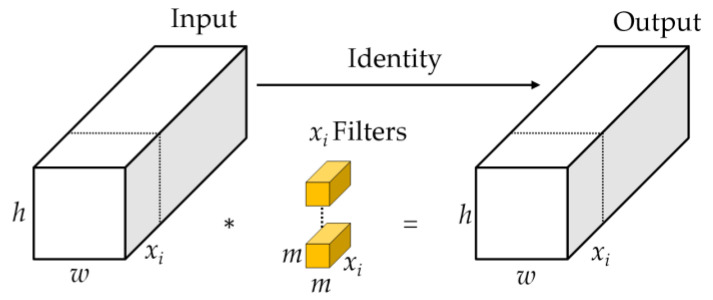
Working principle of PConv. * denotes spatial feature extraction.

**Figure 3 sensors-24-02667-f003:**
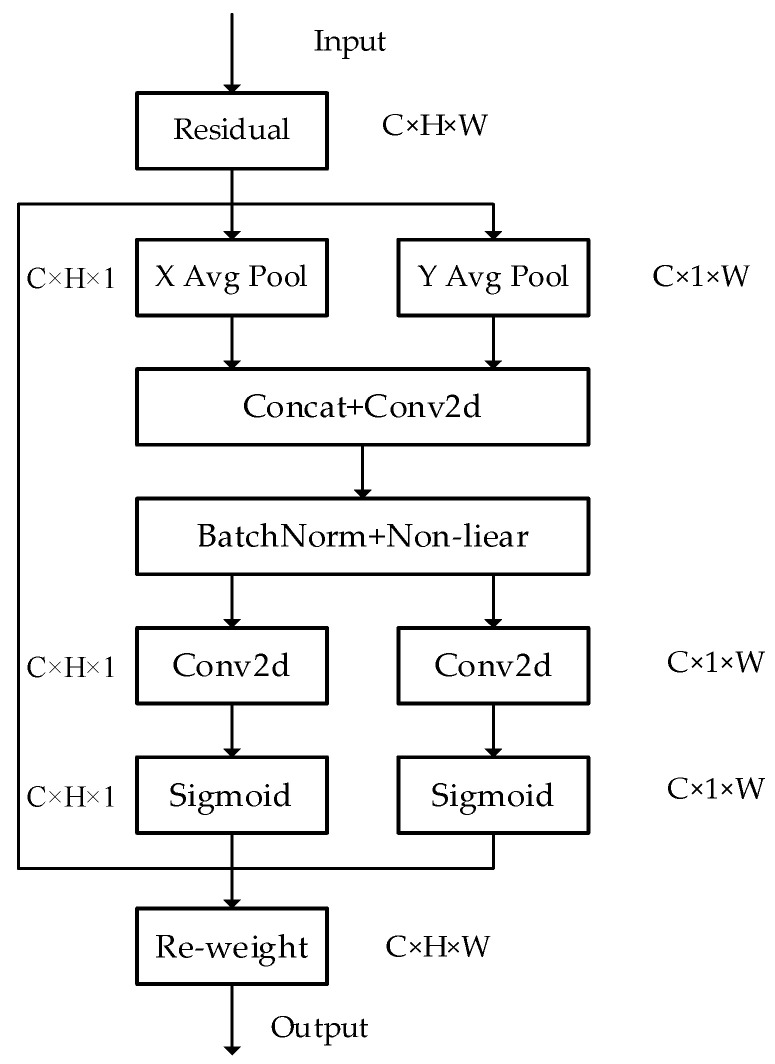
CA module.

**Figure 4 sensors-24-02667-f004:**
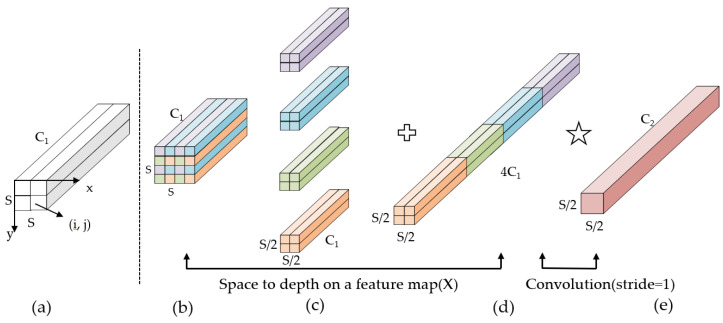
Working principle of SPD convolution module. (**a**) denotes the original feature map; (**b**) denotes the spatial-to-depth transformation; (**c**) denotes channel concatenation; (**d**) denotes an addition operation; (**e**) represents non-strided convolution.

**Figure 5 sensors-24-02667-f005:**
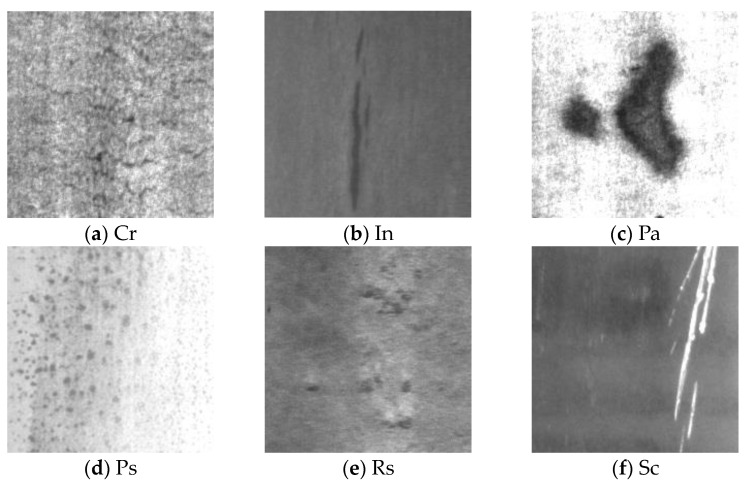
Six defects in NEU-DET dataset.

**Figure 6 sensors-24-02667-f006:**
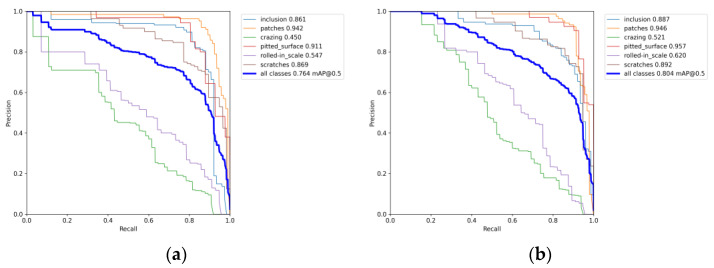
Comparison chart of training results. (**a**) P-R curve of the original YOLOv7; (**b**) P-R curve of the improved YOLOv7.

**Figure 7 sensors-24-02667-f007:**
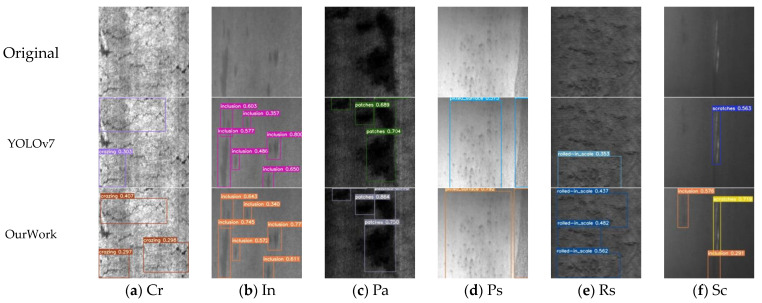
Comparison of Improved Algorithm Effects. (**a**) Comparison of detection effectiveness for Cr-type defects; (**b**) Comparison of detection effectiveness for In-type defects; (**c**) Comparison of detection effectiveness for Pa-type defects; (**d**) Comparison of detection effectiveness for Ps-type defects; (**e**) Comparison of detection effectiveness for Rs-type defects; (**f**) Comparison of detection effectiveness for Sc-type defects.

**Table 1 sensors-24-02667-t001:** Results of Ablation Experiment.

Base	PConv	CA	SPD	mAP (%)	FPS	Par (Mb)	GLOPs
√ ^1^				76.4	92.6	37.2	105.2
√	√			78.6	99.0	32.7	84.9
√	√	√		79.3	95.2	32.9	85.3
√	√	√	√	80.4	90.9	33.9	82.2

^1^ √ signifies the utilization of this algorithm.

**Table 2 sensors-24-02667-t002:** Comparison of detection effects of different attention mechanism modules.

Algorithm	mAP (%)	Cr (%)	In (%)	Pa (%)	Ps (%)	Rs (%)	Sc (%)
Yolov7	76.4	45.0	86.1	94.2	91.1	54.7	86.9
Yolov7 + SE	77.1	46.5	87.8	96.2	90.1	52.7	89.5
Yolov7 + CBAM	77.8	49.2	86.0	94.6	90.5	57.4	89.4
Yolov7 + CA	78.1	49.7	85.8	95.1	89.4	57.6	91.2

**Table 3 sensors-24-02667-t003:** Comparison of detection effects of different IoU.

Algorithm	mAP (%)	Cr (%)	In (%)	Pa (%)	Ps (%)	Rs (%)	Sc (%)
OurWork + CIoU	79.3	52.4	88.0	93.6	93.6	62.9	85.5
OurWork + SIoU	79.6	59.1	88.5	93.5	89.8	58.3	88.0
OurWork + WIoU	79.5	57.0	88.5	95.3	91.9	58.2	86.1
OurWork + EIoU	80.4	52.1	88.7	94.6	95.7	62.0	89.2

## Data Availability

Data are contained within the article.

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
