# Peer review of "Research on Surface Defect Detection of Strip Steel Based on Improved YOLOv7"

_sensors, 2024, doi:10.3390/s24092667_

Round 1

Reviewer 1 Report

Comments and Suggestions for Authors

The paper is well written and auhor proposed to integrate Pconv, coordinate attention and SPD to imporve yolov7's ability to detect surface defect of strip steel band. The author conducted comprehensive experiments and ablation studies to demonstrate the effefectiveness of different modules. It would be good show comparions against two-stage detectors and show the inference time comparisons. 

Author Response

Dear Reviewer

We sincerely appreciate your thorough review and valuable comments on our paper.

The principal objective of this study centers on achieving rapid and accurate detection of surface defects on strip steel. As evidenced by past research experiences, it has become clear that two-stage detection algorithms do not meet the real-time requirements for such inspections, especially when taking into account the limitations of available resources in practical applications. Consequently, with an emphasis on operational efficiency, we have judiciously chosen the one-stage YOLOv7 detection algorithm as the foundation for our work. Given the constraints of existing resources and the necessity for real-time performance, we have proceeded to enhance this base model by incorporating PConv, coordinate attention, and SPD mechanisms, aiming to significantly improve YOLOv7's capability in detecting surface defects on strip steel.

Regarding your suggestion to include comparative experiments with two-stage detectors, while we understand its importance, regrettably, conducting such direct comparisons was beyond the scope of our current experimental setup due to resource constraints and the strict need for real-time responsiveness. Nonetheless, in our subsequent revisions, we will further elaborate on why adopting and improving YOLOv7 is a strategic choice in the third paragraph of the Introduction, and provide theoretical reasoning on the potential superiority of one-stage detection algorithms over two-stage detectors in real-time performance and computational efficiency.

Once again, thank you for your valuable feedback. We look forward to refining our research based on your guidance, ensuring a more compelling presentation in the revised version.

Yours sincerely!

Associate Professor: Yeming zhang

School of mechanical and power engineering, Henan University of Technology

E-mail: zym@hpu.edu.cn

Reviewer 2 Report

Comments and Suggestions for Authors

In this manuscript, a real-time and high-precision defect detection algorithm is proposed to solve the problem that it is difficult to detect strip surface defects quickly at present. The algorithm is based on YOLOv7 and improve the detection performance of steel surface defect small target. This article has clear logic, clear expression and innovative method, but some details need to be improved. It is agreed to be published in this journal after revision.

The specific problems are as follows:

1. The Introduction part of the article should refer to some newly published literature contents for the research status of defect detection. The following article is related research content, and the author can make appropriate reference:

Materials & Design. 237 (2024) 112600ï¼›Journal of Materials Processing Technology. 319 (2023) 118058ï¼›International Journal of Refractory Metals and Hard Materials. 118 (2024) 106460ï¼›Measurement. 224 (2024) 113895ï¼›Engineering 36 (2021) 656–674.

2. The font size in Figure 4 are too small, and the accuracy of recognition results in Figure 6 is small, which is difficult to distinguish.

3. In this paper, the defect detection results are good, but for some images with more background noise, whether the author should consider adding image preprocessing to reduce the background noise.

4. YOLOv7 network has a lot of parameter information, and the time cost is one of the problems to be considered in model training and recognition. Whether the author considers adding lightweight design method to reduce the size of the model.

5. In order to evaluate the model performance more comprehensively, we should add PR curve, confusion matrix and so on to evaluate the model performance.

Author Response

Dear Reviewer

We sincerely appreciate your thorough review and valuable comments on our paper. In response to your suggestions, we fully comprehend and highly regard them. The following is a response to the feedback you provided.

Question 1: The Introduction part of the article should refer to some newly published literature contents for the research status of defect detection. The following article is related research content, and the author can make appropriate reference:

Materials & Design. 237 (2024) 112600ï¼›Journal of Materials Processing Technology. 319 (2023) 118058ï¼›International Journal of Refractory Metals and Hard Materials. 118 (2024) 106460ï¼›Measurement. 224 (2024) 113895ï¼›Engineering 36 (2021) 656–674.4

Answer 1: I fully agree and accept your opinion on updating recent research findings in the introduction section, in order to more comprehensively reflect the latest research trends in the field of defect detection. Based on your recommendation, I have reviewed the following recently published articles and will cite some of them in the revised manuscript to supplement and improve the relevant research status.The object detection algorithms used in the two papers published in Materials&Design and Measurement have certain guiding significance for this article. I have introduced the application results of defect detection techniques discussed in the literature in their respective fields in the third and fourth paragraphs of the introduction, ensuring that readers can have a clear understanding of the current research frontier and technological level.

Question 2: The font size in Figure 4 are too small, and the accuracy of recognition results in Figure 6 is small, which is difficult to distinguish.

Answer 2: To address this issue, we have implemented a moderate enlargement of the font size in Figure 4. Concurrently, in Figure 6, we have refined the accuracy of the recognition results to three decimal places, aiming to enhance the degree of differentiation between these results and thereby improve their distinguishability.

Question 3: In this paper, the defect detection results are good, but for some images with more background noise, whether the author should consider adding image preprocessing to reduce the background noise.

Answer 3: Firstly, we fully agree with your viewpoint. In fact, during the experiment, we did notice that in some complex backgrounds, the presence of noise may affect the accuracy of the detection results. In the preliminary work, we attempted to add CLAHE image preprocessing algorithm to the surface defect dataset of the strip steel, and achieved certain improvements in Sc defect detection. However, we did not achieve satisfactory results in Cr, Pa, and Ps defect detection. Therefore, this article did not conduct ablation experiments on improved algorithms based on image preprocessing using the CLAHE algorithm. Based on your suggestion, we further realize that considering only one preprocessing algorithm has certain limitations. Next, we will verify the impact of other preprocessing methods on the defect detection performance of images containing background noise to ensure that our research can maintain good generalization ability and robustness in various complex scenarios

Question 4: YOLOv7 network has a lot of parameter information, and the time cost is one of the problems to be considered in model training and recognition. Whether the author considers adding lightweight design method to reduce the size of the model.

Answer 4: The question you raised is highly targeted and forward-looking. During the research process, we did realize that the YOLOv7 network may have long training and inference times due to its large number of parameters. To address this challenge, we did have the idea of exploring lightweight model design solutions. As mentioned in the article, we replaced some conventional convolutions in the backbone network with PConv and made the entire network model somewhat lightweight, reducing the number of network parameters by 8.9% compared to the baseline network. In addition, in our previous research, we also introduced the Silmneck structure into the head network to reduce the number of model parameters (reaching 27.4Mb Par, a decrease of about 26.46%). However, the mAP of the detection results decreased slightly, making it difficult to meet the requirements of detection accuracy, and it was ultimately not adopted. Given your suggestion, we feel that it is also necessary to conduct in-depth research on how to combine lightweight technologies such as pruning and knowledge distillation to optimize the YOLOv7 network in future work, in order to reduce model size and runtime while maintaining detection accuracy

Question 5: In order to evaluate the model performance more comprehensively, we should add PR curve, confusion matrix and so on to evaluate the model performance.

Answer 5: We fully understand and agree with the importance of this viewpoint. These evaluation indicators can reflect the predictive accuracy and recall of the model from different dimensions, which helps us to analyze the performance of the model on various samples in more detail. After receiving your feedback, we have decided to include relevant analysis of the PR curve in the article to visually demonstrate the accuracy and recall changes of the model under different thresholds, helping us find the best balance point.

The above is a detailed response to your question, and we sincerely appreciate your valuable insights. We highly value and carefully adopt your suggestions, and will revise and improve the research content based on your guidance, aiming to make our research results more convincing and complete. Thank you again for your professional guidance, which undoubtedly adds indispensable value to our research. We will be committed to continuously improving the quality and reliability of our research.

Yours sincerely

Associate Professor: Yeming zhang

School of mechanical and power engineering, Henan University of Technology

E-mail: zym@hpu.edu.cn

Reviewer 3 Report

Comments and Suggestions for Authors

The contributions of the paper are significant, as it offers a practical solution to a pressing issue in the steel manufacturing industry. By improving detection efficiency, reducing computational overhead, and maintaining real-time performance, the proposed algorithm can potentially enhance production quality and streamline manufacturing processes.

Overall, the paper makes a valuable contribution to the field of industrial automation and computer vision, and its findings are likely to be of interest to researchers and practitioners alike.

Author Response

Dear Reviewer

    We express our sincere gratitude for your positive evaluation and perceptive insights regarding the contributions of our paper. You acknowledged that the paper presents a significant practical solution to a pressing issue in the steel manufacturing industry. By enhancing detection efficiency, minimizing computational overhead, and maintaining real-time performance, the proposed algorithm has the potential to greatly improve production quality and streamline manufacturing processes. We appreciate your recognition that the findings of our study are likely to pique the interest of both researchers and practitioners alike.

Thank you once again for your constructive review and valuable advice on our paper. Your endorsement encourages us to continue our in-depth exploration and strive to develop more effective solutions applicable to industrial practices. We shall take your suggestions into consideration and refine our research further to maximize its impact both theoretically and practically.

Yours sincerely

Associate Professor: Yeming zhang

School of mechanical and power engineering, Henan University of Technology

E-mail: zym@hpu.edu.cn

Reviewer 4 Report

Comments and Suggestions for Authors

Authors: Baozhan Lv, Beiyang Duan, Yeming Zhang, Shuping Li, Feng Wei, Sanpeng Gong, Qiji Ma

 Strip steel can be used to manufacture building materials, electrical components, automotive parts, etc. For such important purposes, steel should have specified chemical and mechanical properties. Today there are a number of strip steel surface-defect detection approaches with different values of mAP, Precision and FPS. However, all these methods need further development to improve defect classification and defect localization precision. In this paper, the real-time defect detection algorithm based on YOLOv7 has been proposed. The strategy demonstrates a number of advantages over the conventional methods. The results and discussion section is logical consistency. In my opinion, the paper can be recommended for publication with minor revisions.

 1.   Abstract, lines 22-23:

The detection speed reaches 90.9 FPS, which effectively improves the detection efficiency of surface defects in strip steel”.

The improved PP-YOLOE-m network reached detection performance of 95 FPS for detecting strip-steel surface defects (Y. Zhang et al. Electronics 2022, 11, 2603).

 2.       Pages 9-10, lines 296-310:

Five evaluation indicators (mAP, Precision, FPS, Params, and FLOPs) were determined in the study. It seems important to estimate Recall using the formula:

Recall=TP / (TP+FP).

The Precision–Recall curves help to compare the performance of the models.

 3.       Page 11, Table 3:

The detection performance of all four loss functions on Cr defects is poor (52,1% - 59,1%).

How can it be explained?

 4. Page 12, line 402:

90.9fps   >   90.9 FPS

Author Response

Dear Reviewer

We sincerely appreciate your thorough review and valuable comments on our paper. In response to your suggestions, we fully comprehend and highly regard them. The following is a response to the feedback you provided.

Answer

 Question 1: Abstract, lines 22-23:“The detection speed reaches 90.9 FPS, which effectively improves the detection efficiency of surface defects in strip steel”.

The improved PP-YOLOE-m network reached detection performance of 95 FPS for detecting strip-steel surface defects (Y. Zhang et al. Electronics 2022, 11, 2603).

Answer 1: Thank you very much for your valuable suggestions on the abstract section of our paper. You pointed out that the detection speed mentioned in lines 22-23 of the abstract is 90.9 frames per second (FPS), and provided a reference for PP-YOLOE-m network to achieve 95FPS detection performance in detecting surface defects on strip steel (Y. Zhang et al., Electronics 2022112603). This may be due to the hardware used for training (NVIDIA GeForce RTX 3070 GPU in this article and Tesla V100 GPU used by Y. Zhang et al.) or different parameter settings.

Based on your suggestion, I deeply realize that the abstract expression in this article is inappropriate. Therefore, I have modified the expression in the abstract of this article by changing "the detection speed reaches 90.9 FPS, effectively improving the detection efficiency of surface defects on strip steel" to "the detection speed reaches 90.9 FPS, which can well meet the requirements of real-time detection".

It is necessary for us to cite the literature in the fourth paragraph of the Introduction section of the article. Once more, I extend my heartfelt appreciation for your invaluable feedback.

Question 2: Pages 9-10, lines 296-310: Five evaluation indicators (mAP, Precision, FPS, Params, and FLOPs) were determined in the study. It seems important to estimate Recall using the formula:

Recall=TP / (TP+FN). The Precision–Recall curves help to compare the performance of the models.

Answer 2: In response to your valuable feedback, we have added the important evaluation indicator of recall rate and presented the model performance more comprehensively and intuitively by drawing Precision–Recall curves.

 Question 3:  Page 11, Table 3: The detection performance of all four loss functions on Cr defects is poor (52,1% - 59,1%). How can it be explained?

Answer 3: We are deeply concerned about the poor performance of all four loss functions in Cr defect detection (52.1% to 59.1%) that you have pointed out, and we provide a possible explanation here: firstly, the inherent complexity and diversity of Cr defects make it difficult for their key features to be fully captured by existing loss functions, especially in areas with fine structures and low contrast, which directly affects the accuracy of detection. Secondly, the problem with training datasets may also be an important factor. If there are insufficient data samples or imbalanced class distribution for Cr defects, it can lead to insufficient model learning, limit its generalization ability, and thus affect the actual detection results. Finally, despite the use of multiple loss functions, an ideal balance has not been achieved between positioning accuracy and category discrimination accuracy. Some loss functions may be too biased towards overall classification performance and overlook precise regression of bounding box positions, and vice versa. Based on the above reasons, the unsatisfactory performance of Cr defect detection has been achieved.

Question 4: Page 12, line 402: 90.9fps   >   90.9 FPS

Answer 4: We have followed your suggestion and corrected "90.9fps" to "90.9 FPS" in the article.

Thank you once again for your constructive review and valuable advice on our paper. Your endorsement encourages us to continue our in-depth exploration and strive to develop more effective solutions applicable to industrial practices. We shall take your suggestions into consideration and refine our research further to maximize its impact both theoretically and practically.

Yours sincerely

Associate Professor: Yeming zhang

School of mechanical and power engineering, Henan University of Technology

E-mail: zym@hpu.edu.cn
